# Insect Cyborgs: Bio-mimetic Feature Generators Improve ML Accuracy on Limited Data

**Charles B. Delahunt[1,2], J. Nathan Kutz[1]**
[1]Applied Math, University of Washington, Seattle, WA
[2]Computational Neuroscience Center, UW, Seattle, WA
{delahunt,kutz}@uw.edu

## Abstract

We seek to auto-generate stronger input features for ML methods faced with limited training data. Biological neural nets (BNNs) excel at fast learning, implying that they extract highly informative features. In particular, the insect olfactory network learns new odors very rapidly, by means of three key elements: A competitive inhibition layer; randomized, sparse connectivity into a high-dimensional sparse plastic layer; and Hebbian updates of synaptic weights. In this work we deploy MothNet, a computational model of the moth olfactory network, as an automatic feature generator. Attached as a front-end pre-processor, MothNet's readout neurons provide new features, derived from the original features, for use by standard ML classifiers. These "insect cyborgs" (part BNN and part ML method) have significantly better performance than baseline ML methods alone on vectorized MNIST and Omniglot data sets, reducing test set error averages 20% to 55%. The MothNet feature generator also substantially out-performs other feature generating methods including PCA, PLS, and NNs. These results highlight the potential value of BNN-inspired feature generators in the ML context.

## 1 Introduction

Machine learning (ML) methods, especially neural nets (NNs) with backprop, often require large amounts of training data to attain their high performance. This creates bottlenecks to deployment, and constrains the types of problems that can be addressed [1]. The limited-data constraint is common for ML targets that use medical, scientific, or field-collected data, as well as AI efforts focused on rapid learning. We seek to improve ML methods' ability to learn from limited data by means of an architecure that automatically generates, from existing features, a new set of class-separating features.

Biological neural nets (BNNs) are able to learn rapidly, even from just a few samples. Assuming that rapid learning requires effective ways to separate classes, we may look to BNNs for effective feature-generators [2]. One of the simplest BNNs that can learn is the insect olfactory network [3], containing the Antennal Lobe (AL) [4] and Mushroom Body(MB) [5], which can learn a new odor given just a few exposures. This simple but effective feedforward network contains three key elements that are ubiquitous in BNN designs: Competitive inhibition [6], high-dimensional sparse layers [7; 8], and a Hebbian update mechanism [9]. Synaptic connections are largely random [10].

MothNet is a computational model of the *M. sexta* moth AL-MB that demonstrated rapid learning of vectorized MNIST digits, with performance superior to standard ML methods given $N \leq 10$ training samples per class [11]. The MothNet model includes three key elements, as follows. (i) Competitive inhibition in the AL: Each neural unit in the AL receives input from one feature, and outputs not only a feedforward excitatory signal to the MB, but also an inhibitory signal to other neural units in the AL that tries to dampen other features' presence in the sample's output AL signature. (ii) Sparsity in the MB, of two types: The projections from the AL to the MB are non-dense ($\approx 15\%$

33rd Conference on Neural Information Processing Systems (NeurIPS 2019), Vancouver, Canada.

non-zero), and the MB neurons fire sparsely in the sense that only the strongest 5% to 15% of the total population are allowed to fire (through a mechanism of global inhibition). (iii) Weight updates affect only MB→Readout connections (AL connections are not plastic). Hebbian updates occur as: $\Delta w_{ij} = \alpha f_i f_j$ if $f_i f_j > 0$ (growth), and $\Delta w_{ij} = -\delta w_{ij}$ if $f_i f_j = 0$ (decay), where $f_i, f_j$ are two neural firing rates ($f_i \in$ MB, $f_j \in$ Readouts) with connection weight $w_{ij}$.

In this work we tested whether the MothNet architecture can usefully serve as a front-end feature generator for an ML classifier (our thanks to Blake Richards for this suggestion). We combined MothNet with a downstream ML module, so that the Readouts of the trained AL-MB model were fed into the ML module as additional features. From the ML perspective, the AL-MB acted as an automatic feature generator; from the biological perspective, the ML module stood in for the downstream processing in more complex BNNs. Our Test Case was a non-spatial, 85-feature, 10-class task derived from the downsampled, vectorized MNIST data set (hereafter "*v*MNIST"). On this non-spatial dataset, CNNs or other spatial methods were not applicable.

The trained Mothnet Readouts, used as features, significantly improved the accuracies of ML methods (NN, SVM, and Nearest Neighbors) on the test set in almost every case. That is, the original input features (pixels) contained class-relevant information unavailable to the ML methods alone, but which the AL-MB network encoded in a form that enabled the ML methods to access it. MothNet-generated features also significantly out-performed features generated by PCA (Principal Components Analysis), PLS (Partial Least Squares), NNs, and transfer learning (weight pretraining) in terms of their ability to improve ML accuracy. These results indicate that the insect-derived network generated significantly stronger features than these other methods.

## 2 Experimental setup

To generate *v*MNIST, we downsampled, preprocessed, and vectorized the MNIST data set to give samples with 85 pixels-as-features. *v*MNIST has the advantage that our baseline ML methods (Nearest Neighbors, SVM, and Neural Net) do not attain full accuracy at low *N*. Trained accuracy of baseline ML methods was controlled by restricting training data. Full network architecture details of the AL-MB model (MothNet) are given in [11]. Full Matlab code for these cyborg experiments including comparison methods, all details re ML methods and hyperparameters, and code for MothNet simulations, can be found at [12]. MothNet instances were generated randomly from templates that specified connectivity parameters. We ran two sets of experiments:

**Cyborg vs baseline ML methods on *v*MNIST** Experiments were structured as follows:
  1. A random set of *N* training samples per class was drawn from *v*MNIST.
  2. The ML methods trained on these samples, to provide a baseline.
  3. MothNet was trained on these same samples, using time-evolved stochastic differential equation simulations and Hebbian updates, as in [11].
  4. The ML methods were then retrained from scratch, with the Readout Neuron outputs from the trained MothNet instance fed in as additional features. These were the "insect cyborgs", *i.e.* an AL-MB feature generator joined to a ML classifier.
  5. Trained ML accuracies of the baselines and cyborgs were compared to assess gains.

**MothNet features vs other feature generators** To compare the effectiveness of MothNet features vs features generated by conventional ML methods, we ran *v*MNIST experiments structured as as above, but with MothNet replaced by one of the following feature generators:
  1. PCA applied to the *v*MNIST training samples. The new features were the projections onto each of the top 10 modes.
  2. PLS applied to the *v*MNIST training samples. The new features were the projections onto each of the top 10 modes. Since PLS incorporates class information, we expected it to out-perform PCA.
  3. NN pre-trained on the *v*MNIST training samples. The new features were the (logs of the) 10 output units. This feature generator was used as a front end to SVM and Nearest Neighbors only. Since *v*MNIST has no spatial content, CNNs were not used.
  4. NN with weights initialized by training on an 85-feature vectorized Omniglot data set [13], then trained on the *v*MNIST data as usual (transfer learning, applied to the NN baseline only). Omniglot is an MNIST-like thumbnail collection of 1623 characters with 20 samples each.
  (5.) For the baseline NN method, we used one hidden layer. Including two hidden layers did not

improve baseline performance. This was an implicit control, showing that MothNet features were not equivalent to just adding an extra layer to a NN.

## 3  Results

MothNet readouts as features significantly improved accuracy of ML methods, demonstrating that the MothNet architecture effectively captured new class-relevant features. We also tested a non-spatial, 10-class task derived from the Omniglot data set and found similar gains. MothNet-generated features were also far more effective than the comparison feature generators (PCA, PLS, and NN).

**Gains due to MothNet features on *v*MNIST**
ML baseline test set accuracies ranged from 10% to 88%, depending on method and on $N$ (we stopped our sweep at $N = 100$). This baseline accuracy is marked by the lower colored circles in Fig 1. Cyborg test set accuracy is marked by the upper colored circles in Fig 1, and the raw gains in accuracy due to MothNet features are marked by thick vertical bars. MothNet features increased raw accuracy across all ML models. Relative reduction in test set error, as a percentage of baseline error, was 20% to 55%, with high baseline accuracies seeing the most benefit (Fig 2). NN models saw the greatest benefits, with 40% to 55% relative reduction in test error. Remarkably, a MothNet front-end improved ML accuracy even in cases where the ML baseline already exceeded the $\approx 75\%$ accuracy ceiling of MothNet (*e.g.* NNs at $N$ = 15 to 100 samples per class): the MothNet readouts contained clustering information which ML methods leveraged more effectively than MothNet itself. Gains were significant in almost all cases with $N > 3$. Table 1 gives $p$-values of the gains due to MothNet.

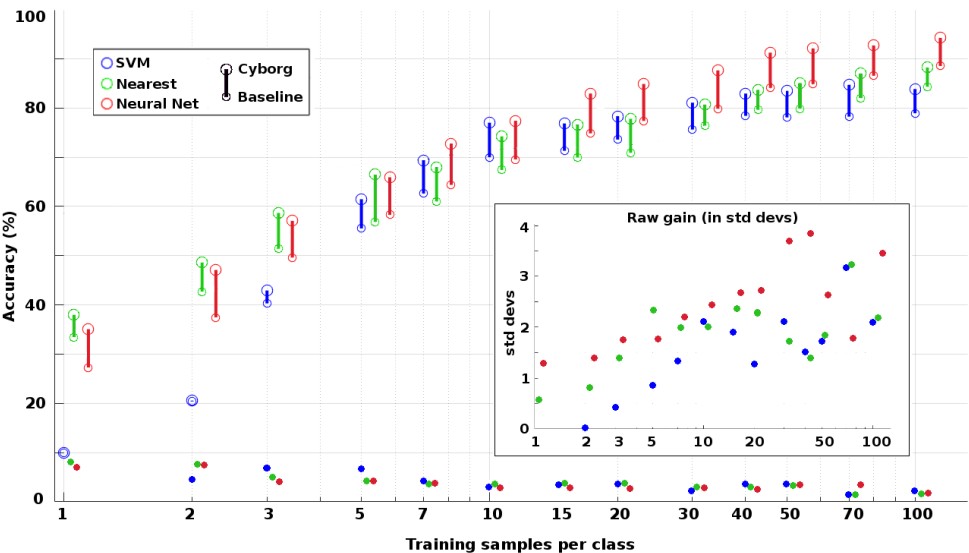

Figure 1: Test set accuracy of baseline ML and cyborg classifiers, vs $N$ training samples per class. Baseline ML accuracies are shown as small circles, cyborg accuracies are shown as larger circles, and thick vertical bars mark the increase in accuracy. MothNet test accuracy is $\approx 65\%$ to 75%. Baseline methods' std dev ($\sigma$) are given as solid dots near the x-axis. Inset: Gain in accuracy (cyborg over ML baseline), in units of std dev (Fisher discriminant), showing that gains were significant (see Table 1).

Table 1: $p$-values of gains over ML baseline due to MothNet. Gains were significant for all $N > 3$.

| Method | $N$ =1 | 2 | 3 | 5 | 7 | 10 | 15 | 20 | 30 | 50 | 70 | 100 |
|---|---|---|---|---|---|---|---|---|---|---|---|---|
| NearNeigh | .58 | .42 | .20 | .02 | .04 | .04 | .02 | .01 | .09 | .07 | .00 | .03 |
| SVM | 1.00 | .96 | .31 | .39 | .18 | .04 | .06 | .16 | .04 | .08 | .00 | .04 |
| Neural Net | .89 | .76 | .48 | .07 | .03 | .01 | .01 | .01 | .00 | .01 | .08 | .00 |

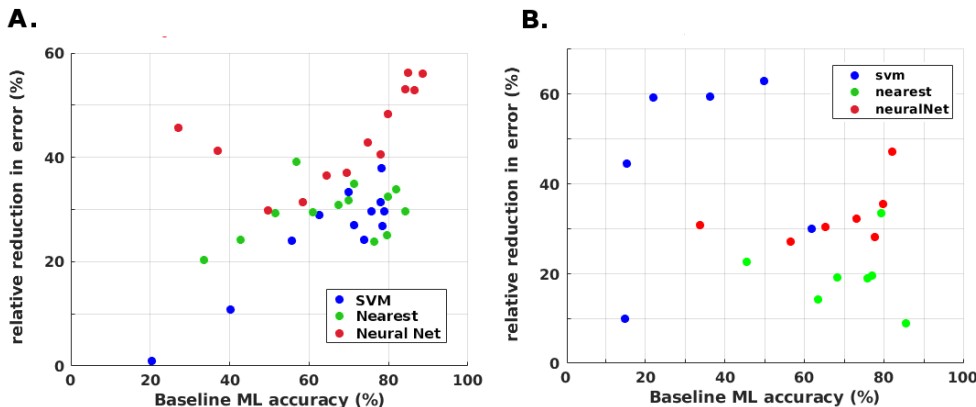

Figure 2:   Relative reduction in test set error due to MothNet features. **A:** On *v*MNIST, error reductions were especially high (40% to 60%) for NNs. **B:** On *v*Omniglot error reductions were highest for NNs and SVMs.

**Comparison to other feature generators**
We ran the cyborg framework on *v*MNIST using PCA (projections onto top 10 modes), PLS (projection onto top 10 modes), and NN (logs of the 10 output units) as feature generators. Each feature generator was trained (*e.g.* PCA projections were defined) using the training samples. Table 2 gives the relative increase in mean accuracy due to the various feature generators (or to pre-training) for NN models (13 runs per data point). Results for Nearest Neighbors and SVM were similar. MothNet features were far more effective than these other methods.

Table 2: Mean relative percentage increase in accuracy due to various feature generators ("F Gen"), for NN classifiers. "preTrain " means: pretrain weights on Omniglot, then train on *v*MNIST.

| **F Gen** | $N$=1 | 2 | 3 | 5 | 7 | 10 | 15 | 20 | 30 | 50 | 70 | 100 |
|---|---|---|---|---|---|---|---|---|---|---|---|---|
| PCA | -57 | 0.2 | -0.8 | 1.2 | 2.6 | 1.7 | 0.3 | 1.3 | -0.3 | 0.2 | 0.3 | 0.2 |
| PLS | NA | 0.2 | 5.9 | 1.0 | 1.5 | 2.8 | -0.2 | 1.2 | 0.3 | 1.6 | 1.5 | 1.9 |
| preTrain | **15** | 4.2 | 5.8 | -3.1 | -1.1 | 0.2 | 1.3 | 1.5 | -3.4 | -0.4 | -4.7 | -1.1 |
| MothNet | 4 | **17** | **15** | **13.1** | **13** | **11.3** | **10.8** | **9.0** | **9.7** | **8.5** | **7.1** | **6.4** |

**Effect of pass-through AL**
The MothNet architecture has two main layers: a competitive inhibition layer (AL) and a high-dimensional, sparse layer (MB). To test the effectiveness the MB alone, we ran the *v*MNIST experiments, but using a pass-through (identity) AL layer for MothNet. Cyborgs with a pass-through AL still posted significant improvements in accuracy over baseline ML methods. The gains of cyborgs with pass-through ALs were generally between 60% and 100% of the gains posted by cyborgs with normal ALs (see Table 3), suggesting that the high-dimensional, trainable layer (the MB) was most important. However, the competitive inhibition of the AL layer clearly added value in terms of generating strong features, up to 40% of the total gain. NNs benefitted most from the AL layer.

Table 3: Relative importance of the MB, vs number of training samples per class $N$. Entries give the gains posted by cyborgs with pass-through ALs as a percentage of the gains of full cyborgs, for the three ML methods. Entries = 100% indicate that average gains from the pass-through AL were $\geq$ average gains from the normal AL.

| **Method** | $N$ =1 | 2 | 3 | 5 | 7 | 10 | 15 | 20 | 30 | 40 | 50 | 70 | 100 |
|---|---|---|---|---|---|---|---|---|---|---|---|---|---|
| NearNeigh | 82 | 100 | 91 | 76 | 100 | 100 | 58 | 74 | 88 | 64 | 100 | 100 | 65 |
| SVM | NA | NA | 100 | 87 | 79 | 97 | 75 | 94 | 98 | 82 | 100 | 76 | 15 |
| NN | 100 | 60 | 62 | 67 | 75 | 91 | 100 | 93 | 100 | 100 | 100 | 82 | 65 |

## 4 Discussion

We deployed an automated feature generator based on a very simple BNN, containing three key elements rare in engineered NNs but endemic in BNNs of all complexity levels: (i) competitive inhibition; (ii) sparse projection into a high-dimensional sparse layer; and (iii) Hebbian weight updates for training. This bio-mimetic feature generator significantly improved the learning abilities of standard ML methods on both *v*MNIST and *v*Omniglot. Class-relevant information in the raw feature distributions, not extracted by the ML methods alone, was evidently made accessible by MothNet's pre-processing. In addition, MothNet features were consistently much more useful than features generated by standard methods such as PCA, PLS, NNs, and pre-training.

The competitive inhibition layer may enhance classification by creating several attractor basins for inputs, each focused on the features that present most strongly for a given class. This may push otherwise similar samples (of different classes) away from each other, towards their respective class attractors, increasing the effective distance between the samples. The sparse connectivity from AL to MB has been analysed as an additive function, which has computational and anti-noise benefits [14].

The insect MB brings to mind sparse autoencoders (SAs) *e.g.* [15]. However, there are several differences: MBs do not seek to match the identity function; the sparse layers of SAs have fewer active neurons than the input dimension, while in the MB the number of active neurons is much greater than the input dimension; MBs have no pre-training step; and the MB needs very few samples to bake in structure that improves classification. The MB differs from Reservoir Networks [16] in that MB neurons have no recurrent connections.

Finally, the Hebbian update mechanism appears to be quite distinct from backprop. It has no objective function or output-based loss that is pushed back through the network, and Hebbian weight updates, either growth or decay, occur on a local "use it or lose it" basis. We suspect that the dissimilarity of the optimizers (MothNet vs ML) was an asset in terms of increasing total encoded information.

**Acknowledgements**
Our thanks to Blake Richards, who suggested these experiments.
CBD's work was partially supported by the Swartz Foundation.
JNK acknowledges support from the Air Force Office of Scientific Research (FA9550-19-1-0011).

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
