# OpenReview forum: "Insect Cyborgs: Bio-mimetic Feature Generators Improve ML Accuracy  on Limited Data"
_NeurIPS.cc/2019/Workshop/Neuro_AI — Real Neurons & Hidden Units @ NeurIPS 2019 Poster_

### Official Review · AnonReviewer3 · 2019-09-26
**feature extraction method based on neural network simulated from olfactory structure**

**Clarity:** 3

**Comment:**

Good as preliminary, but need significant improvement.

**Category:**

Neuro->AI

**Clarity Comment:**

The paper is clearly written, but missing some critical control to show the advantage of this feature extraction network.

**Evaluation:**

3: Good

**Importance:**

3: Important

**Importance Comment:**

This paper build a biologically inspired neural network from the moth olfactory system and used that as a feature extraction network to preprocess images before using other machine learning algorithms. This is an interesting idea.

**Intersection:**

3: Medium

**Intersection Comment:**

Nevertheless, the biologically inspired network is a neat idea and has great potential.

**Rigor Comment:**

The authors identified key component for computation from the moth olfactory system. They showed using this network to preprocess image pixel data and generate features as input for other machine learning algorithm can increase performance.  However, the preprocessing step is like adding more layers to a CNN (more parameters) and this comparison is not convincing enough to show the importance of the biologically inspired network.

**Technical Rigor:**

2: Marginally convincing

---

### Official Review · AnonReviewer2 · 2019-09-26
**An interesting attempt to use neuroscience to inspire AI**

**Clarity:** 3

**Comment:**

Overall,  I think this is a quite interesting contribution showing some promise of integrating computational principles learned from neuroscience to ML.
A few comments/suggestions.
First, it would be great to see the method tested in more challenging datasets to see if the results generalized.
Second, it would be helpful to gain some insights about why the performance improves. One possible idea-  because there are several ingredients in the MothNet, one could keep a subset of these and see how that change the performance.


**Category:**

Neuro->AI

**Clarity Comment:**

I found the paper is well-written and relatively easy to follow., although the paper would improve if the motivations could be better articulated.

**Evaluation:**

4: Very good

**Importance:**

3: Important

**Importance Comment:**

The paper presents a biologically-insured model for classification, i.e., Cyborg.
The idea of using computational principles in neuroscience to inspire machine learning/AI is an important research direction. I think this paper represents an interesting attempt along this direction. It could help inspiring future endeavors on this topic.

**Intersection:**

3: Medium

**Intersection Comment:**

Although the work heavily relies on MothNet, which has been proposed previously. However, the authors' serious effort to combine the ingredients from neuroscience and AI to come out with better method should be applauded.

**Rigor Comment:**

The proposed method attach a previously proposed model MothNet, which is inspired based on the physiology of insect olfaction system to a ML classifier. The MothNet acts as a front-end feature generator.

The authors compare their method to several baseline methods. They also tried other feature generators rather than MothNet.  Overall, the authors found that Cyborg can achieve better performance on down-sampled, vectorized MNIST, Omnigplot.

The techniques used are solid in general.

**Technical Rigor:**

4: Very convincing

---

### Official Review · AnonReviewer1 · 2019-09-27
**Lack of details and proper motivation renders MothNet useless**

**Clarity:** 3

**Category:**

Not applicable

**Clarity Comment:**

The manuscript is readable but the logic is very unclear.

**Evaluation:**

1: Very poor

**Importance:**

1: Irrelevant

**Importance Comment:**

The relevance of this work was not at all clear.

**Intersection:**

1: Very low

**Intersection Comment:**

Not clear at all what is BNN or how it is relevant. Some claim about faster learning is mentioned. Not sure why.

**Rigor Comment:**

Not clear why they did what they did.

**Technical Rigor:**

2: Marginally convincing

---

### Decision · Program_Chairs · 2019-10-02

Accept (Poster)